# Lamivudine and Emtricitabine Dosing Proposal for Children with HIV and Chronic Kidney Disease, Supported by Physiologically Based Pharmacokinetic Modelling

**DOI:** 10.3390/pharmaceutics15051424

**Published:** 2023-05-06

**Authors:** Tom G. Jacobs, Marika A. de Hoop-Sommen, Thomas Nieuwenstein, Joyce E. M. van der Heijden, Saskia N. de Wildt, David M. Burger, Angela Colbers, Jolien J. M. Freriksen

**Affiliations:** 1Department of Pharmacy, Research Institute for Medical Innovation, Radboud University Medical Center, 6525 GA Nijmegen, The Netherlands; 2Department of Pharmacology and Toxicology, Research Institute for Medical Innovation, Radboud University Medical Center, 6525 GA Nijmegen, The Netherlands; 3Department of Pediatrics, Erasmus MC-Sophia’s Children’s Hospital, 3015 CN Rotterdam, The Netherlands

**Keywords:** HIV, physiologically based pharmacokinetic modelling, chronic kidney disease, paediatrics, emtricitabine, lamivudine

## Abstract

Dose recommendations for lamivudine or emtricitabine in children with HIV and chronic kidney disease (CKD) are absent or not supported by clinical data. Physiologically based pharmacokinetic (PBPK) models have the potential to facilitate dose selection for these drugs in this population. Existing lamivudine and emtricitabine compound models in Simcyp^®^ (v21) were verified in adult populations with and without CKD and in non-CKD paediatric populations. We developed paediatric CKD population models reflecting subjects with a reduced glomerular filtration and tubular secretion, based on extrapolation from adult CKD population models. These models were verified using ganciclovir as a surrogate compound. Then, lamivudine and emtricitabine dosing strategies were simulated in virtual paediatric CKD populations. The compound and paediatric CKD population models were verified successfully (prediction error within 0.5- to 2-fold). The mean AUC ratios in children (GFR-adjusted dose in CKD population/standard dose in population with normal kidney function) were 1.15 and 1.23 for lamivudine, and 1.20 and 1.30 for emtricitabine, with grade-3- and -4-stage CKD, respectively. With the developed paediatric CKD population PBPK models, GFR-adjusted lamivudine and emtricitabine dosages in children with CKD resulted in adequate drug exposure, supporting paediatric GFR-adjusted dosing. Clinical studies are needed to confirm these findings.

## 1. Introduction

In 2020, an estimated 37.7 million people were living with the human immunodeficiency virus (HIV) worldwide, of which 1.7 million were younger than 15 years of age [1,2]. HIV infection has been associated with impaired kidney function in adults and children [3,4,5]. Generally, kidney impairment results from HIV-associated nephropathy, HIV-associated immune complex kidney disease, viral co-infections, or kidney damage induced by antiretroviral drugs [6]. While disease aetiology may differ, all HIV-associated kidney disease may potentially develop into chronic kidney disease (CKD). CKD in children is defined as having signs of structural or functional alterations in the kidney or an estimated glomerular filtration rate (GFR) persisting below 60 mL/min/1.73 m^2^ for at least three months [7]. Various grades of CKD include grade 2 (G2; GFR 60–90 mL/min/1.73 m^2^), grade 3 (G3; GFR 30–60 mL/min/1.73 m^2^), grade 4 (G4; GFR 15–30 mL/min/1.73 m^2^), and grade 5 (G5; GFR < 15 mL/min/1.73 m^2^ [8]), according to the ‘Kidney Disease: Improving Global Outcomes (KDIGO)’ guideline [8]. Before the era of paediatric combination antiretroviral therapy (cART), up to 40% of children living with HIV were reported to suffer from kidney disease [9]. Although the introduction of cART has substantially reduced the prevalence of kidney disease, various studies reported that between 5.1% and 14.9% of children living with HIV, both in resource-rich and resource-limited settings, have an abnormal GFR [6,10]. This high number may be related to the fact that not all children living with HIV receive effective cART (only 52% in 2021) [1,11,12].

cART consists of a combination of antiretroviral drugs inhibiting various steps in the HIV replication cycle. First-line cART regimens for both children and adults typically include either lamivudine (3TC) or emtricitabine (FTC), which are both nucleoside reverse transcriptase inhibitors (NRTIs) [13]. Both 3TC and FTC are approved by regulatory authorities for use in children for the treatment of HIV. 3TC is more commonly used by children globally due to its widespread availability [14,15]. 

Both 3TC and FTC are eliminated renally, primarily in the unchanged form, through glomerular filtration as well as active tubular secretion by organic cation transporter 2 (OCT2) and multidrug and toxin extrusion (MATE) proteins 1 and 2 [16,17,18]. In patients with CKD, GFR as well as tubular secretion are generally reduced as a result of kidney damage, leading to decreased FTC and 3TC clearance and increased drug exposure [17,19,20]. The product labels and international guidelines include recommendations to reduce the dose for adults with GFR <50 mL/min/1.73 m^2^, a reduction in the total daily dose for 3TC and a prolonged dosing interval for FTC [21,22]. These dosing recommendations are based on drug exposure matching with adults with normal kidney function [17,20]. 

The 3TC product label states that, for children with reduced GFR, a similar relative dose reduction can be applied as has been recommended for adults with a reduced GFR [18]. For FTC, guidance on how to adjust the dose for paediatric patients with a reduced GFR is not provided. To our knowledge, the clinical trials that have been performed in which the pharmacokinetics (PK) of 3TC and FTC were explored in children excluded children with CKD, did not assess GFR, or did not specifically analyse differences in PK between children with and without CKD. As such, there are no data available to establish evidence-based dosing recommendations for this population. 

Extrapolating adult dosages to paediatrics is complex as developmental changes and maturation processes of drug absorption, distribution, metabolism, and excretion (e.g., drug transporter ontogeny) should be taken into account [23]. On top of that, many of these processes are affected by CKD, leading to large interpatient variability in drug PK. Extrapolation of dose recommendations for adults with CKD to children with CKD is, therefore, not straightforward and could potentially result in suboptimal dosing.

In the absence of clinical PK data, physiologically based pharmacokinetic (PBPK) modelling can be used to predict PK in virtual special populations and guide decision making with regard to dosing. PBPK modelling uses information on the compound of interest (e.g., physiochemical properties) to predict PK in a set of virtual subjects with user-defined population characteristics (e.g., organ volumes, weight, and kidney function). A user-defined trial (e.g., age range and dosing strategy) can be designed and conducted in silico [24]. 

In this study, we aimed to develop and verify population models in Simcyp^®^ reflecting paediatric subjects with different degrees of CKD and use these models to support extrapolated dose recommendations for 3TC and FTC in children with HIV and CKD.

## 2. Materials and Methods

The PBPK models were developed, and simulations were conducted in Simcyp^®^ (v21, Certara UK Limited, Simcyp Division, Sheffield, UK). As clinical PK data for 3TC and FTC are not available for paediatric subjects with CKD, we used ganciclovir (GCV) as a surrogate compound to verify the novel paediatric CKD population models. Similar to 3TC and FTC, GCV undergoes both glomerular filtration, in similar proportions to 3TC and FTC, and tubular secretion (through organic anion transporter 1 (OAT1) and MATE), and clinical PK data are available for this drug in children with CKD. Compound models of 3TC, FTC, and GCV were derived from the Simcyp^®^ repository and used without modifications [25,26,27]. The GCV compound model is designed and verified for oral administration in the form of valganciclovir, which is the prodrug of GCV. All compound models included a full PBPK distribution model, accommodating the permeability-limited Mechanistic Kidney Model (Mech KiM) to simulate renal drug handling in a mechanistic manner (Appendix A) [25,26,27]. The “Sim-Healthy volunteer”, “Sim-Renal Impaired_Mild” (CKD G2), “Sim-Renal Impaired_Moderate” (CKD G3), “Sim-Renal Impaired_Severe” (CKD G4)—all adult populations—and “Sim-Paediatric” populations from the Simcyp^®^ software were used (Appendix A). 

As 3TC, FTC, and GCV are excreted through glomerular filtration as well as tubular secretion; total clearance of these compounds can be calculated as follows:$${CL}_{total}={CL}_{filt}+{CL}_{tub}+{CL}_{add}$$
where Cl_total_ refers to total clearance, Cl_filt_ refers to glomerular filtration, Cl_tub_ refers to active tubular secretion, and CL_add_ refers to the additional non-mechanistic clearance component.

In this study, we focused on all three components contributing to total clearance. Reduction in CL_filt_ is yet incorporated in the Simcyp^®^ adult CKD population models as a reduction in the GFR, based on published GFR data for the corresponding CKD population. CL_tub_ is also reduced in subjects with CKD. To account for the reduction in CL_tub_, the equation used to calculate individual adult kidney size is adapted in the Simcyp^®^ default “Sim-Renal Impaired_Moderate” and “Sim-Renal Impaired_Severe” populations. The number of proximal tubule cells per gram of kidney (PTCPGK) was not adapted in the default Simcyp^®^ adult CKD population models, although others have shown that a drug-specific reduction in PTCPGK was needed to better reflect the reduction in CL_tub_, as it describes the loss of tubular cells [25,27,28]. Hence, we decided to decrease PTCPGK, which is further specified in Table 1 and Appendix A. Furthermore, CL_add_ is also believed to be reduced in patients with CKD and, hence, we also incorporated a change in CL_add_ based on a linear correlation of CL_add_ to GFR [25]. More details on how CL_filt_, CL_tub_, and CL_add_ are adapted in the CKD population models are provided in Table 1.

Model performance was verified via visual inspection of the predicted and observed plasma concentration–time curves and by comparing the predicted PK parameter values with those observed in clinical PK studies. To adequately carry this out, we designed and conducted virtual trials similar to the PK studies from which the observed data are obtained (e.g., age, proportion of females, and dosing strategy). Verification was deemed successful when the prediction error for the PK parameters (predicted/observed: P/O) was within a range of 0.5 to 2. 

First, PBPK model performance was assessed for predicting the PK of 3TC, FTC, and GCV in adults with normal kidney function, in adults with various degrees of CKD, and in paediatric subjects with normal kidney function (2.1 PBPK model verification simulations—Run 1). Then, we developed the paediatric CKD population models (2.2 Development of paediatric CKD population models), after which the population models were verified using GCV PK data from paediatric subjects with CKD (2.3 PBPK model verification simulations—Run 2). Finally, prospective simulations were conducted to assess 3TC and FTC PK upon standard and GFR-adjusted dosing in virtual paediatric subjects with CKD (2.4 Prospective simulations of 3TC and FTC PK in paediatric and adult CKD populations). The stepwise model verification workflow is shown in Figure 1.

### 2.1. PBPK Model Verification Simulations—Run 1

#### 2.1.1. Adult Population with Normal Kidney Function

The PK of 3TC, FTC, and GCV were simulated in adults with normal kidney function using the default North European white “Sim-Healthy volunteer” population. Predictions of PK were compared to observed PK data reported by Heald et al., Wang et al., and Czock et al. [29,30,31]. 

#### 2.1.2. Adult Populations with Various Degrees of CKD 

We simulated the PK of the three compounds in adult populations with various degrees of CKD to confirm whether the PBPK model was able to adequately predict 3TC, FTC, and GCV exposure in these populations. Final virtual population characteristics, including adjusted parameters, are reported in the Appendix A. Model performance was assessed by comparing predicted PK data with observed PK data of all compounds in different adult CKD populations (Appendix A) [17,20,29,30,31]. 

#### 2.1.3. Paediatric Population with Normal Kidney Function

Model verification of 3TC, FTC, and GCV in a paediatric population with normal kidney function was conducted using the default “Sim-Paediatric” population. Individual 3TC PK data were available from a paediatric clinical study by Burger et al. [32]. Children <25 kg received 4 mg/kg 3TC solution twice daily (BID) and those ≥25 kg a fixed dose of 150 mg 3TC tablets BID. For FTC, clinical data were extracted from two paediatric studies [33,34]. Simulations of FTC PK were conducted for various age groups receiving an oral solution of 120 mg/m^2^ FTC (<2 years; 2 to <6 years; 6 to <12 years) and those receiving a fixed-dose dispersible tablet of 200 mg (6 to <11 years; 12 to 17 years). Predicted PK were compared to the respective observed data. FTC oral solution has a lower bioavailability compared to capsules, which was corrected for by adjusting the fraction of the dose absorbed (fa) in the FTC compound model (Appendix A) [17,34]. Model verification for GCV was assessed with individual GCV PK data from a study by Vaudry et al. These children received an individualized oral dose of valganciclovir based on their body surface area (BSA) and GFR using the Schwartz bedside equation (see Section 2.3) [35]. 

### 2.2. Development of Paediatric CKD Population Models

Since population models reflecting children with CKD are not available in Simcyp^®^ v21, the “Sim-Paediatric” model was adjusted to build these population models. We defined paediatric CKD groups based on GFR values similar to adults CKD populations. Applying the KDIGO GFR ranges for children under two years of age could be inaccurate, as their GFR (corrected for BSA) is typically lower than that of older individuals due to maturation [8]. We applied a scaling factor to define individual GFR values of paediatric subjects from a specific CKD group (Appendix A). This allowed us to use the validated Simcyp^®^ built-in equation to calculate GFR that accounts for kidney maturation in infants. Then, CL_filt_, CL_tub_, and CL_add_ were adjusted based on extrapolation from the adult CKD populations. We employed the paediatric Mech Kim model to account for the effect of age on tubular secretion. The model builds upon the Simcyp^®^ adult Mech Kim model and incorporates transporter ontogeny, as well as age-related variations in anatomical and physiological kidney parameter [36] models. A detailed description can be found in Table 1 and Appendix A.

### 2.3. PBPK Model Verification Simulations—Run 2

Next, we used GCV as a surrogate compound to verify the paediatric CKD population. Predicted AUC values were compared with individual AUC data extracted from a clinical study with GCV [35]. The authors plotted individual AUC values against BSA and BSA-normalized GFR, which enabled us to calculate individually administered valganciclovir doses (valganciclovir dose (mg) = 7 × BSA × GFR, Schwartz bedside method) and subsequently calculate the equivalent GCV doses. Based on BSA, we estimated the age, weight, and length of all individuals, which were then incorporated into our virtual trial design to predict the AUC for each individual. Each individual (age range: 3 months–15 years) was assigned to a specific CKD group, based on BSA-normalized GFR (no CKD: >90, CKD G2: 60–90, CKD G3: 30–60 mL/min/1.73 m^2^). No observed GCV data were available in children with a GFR < 30 mL/min/1.73 m^2^. Subsequently, a virtual trial was conducted consisting of 50 subjects, with the characteristics (i.e., age, weight, GFR, and dose) of one specific individual. This was repeated for all other individuals. Simulated PK parameter values were compared to their respective observed clinical value, and the resulting prediction errors were pooled per CKD group.

### 2.4. Prospective Simulation of 3TC and FTC PK in Paediatric CKD Populations 

The product labels of 3TC and FTC provide weight-based dose recommendations for children weighing less than 25 and 33 kg, respectively, whereas those with a higher weight are recommended to use a fixed dose, see Table 2. These doses are hereafter referred to as the ‘standard doses’. We distinguished various age groups based on the age range for which the paediatric CKD model was verified, the standardized paediatric subpopulations defined by the US FDA, and the age/weight range for different dosing strategies (mg/kg versus a fixed dose): 3 months to <2 years, 2 to <6 years, 6 to <12 years, 12 to <15 years, and adults. For subjects between 15 and 18 years of age, GFR is calculated in a similar manner as compared to adults, which explains the age cut-off of 15 years. First, we simulated PK of 3TC and FTC upon administration of the standard dose in adults and all paediatric age groups, both with normal kidney function as well as with CKD G3 (GFR 30–50 mL/min/1.73 m^2^) and CKD G4 (GFR 15–30 mL/min/1.73 m^2^). Then, PK after GFR-adjusted doses were simulated in adults and all paediatric age groups with CKD G3 and CKD G4. Paediatric GFR-adjusted doses for 3TC were adopted from the product label. For FTC, the standard paediatric dose is reduced proportionally to the adult dose to obtain the GFR-adjusted dose. For the oral solution, this involves a total dose reduction and, for the capsule, the dosing interval was prolonged. The doses are displayed in Table 2. A total of four doses (either standard dose or GFR-adjusted dose) were administered in the virtual trials to reach steady state, after which PK parameter values were determined. To evaluate whether the GFR-adjusted doses in children with CKD led to the same exposure as the standard dose in children with normal kidney function, AUC_0–24h_ and C_max_ ratios (CKD population/no CKD populations) were calculated.

## 3. Results

### 3.1. PBPK Model Verification Simulations—Run 1 

The 3TC, FTC, and GCV compound models (default models, Appendix A) were successfully verified for predicting PK in adults with normal kidney function. CKD group-specific adjustments of parameter values determining CL_filt_, CL_tub_, and CL_add_ seemed accurate as the PBPK models adequately predicted 3TC, FTC, and GCV PK in adults with CKD. Furthermore, the paediatric PBPK model was able to adequately predict PK of 3TC, FTC, and GCV in children with normal kidney function. Model verification results are included in Appendix A. All prediction errors were within 2-fold.

### 3.2. Verification of Paediatric CKD Populations—Run 2

GCV was used as a surrogate compound to verify the developed paediatric CKD population models. For 39 individuals (aged 3 months–15 years), simulations of GCV PK were conducted. Figure 2 includes all individual prediction errors per CKD group; prediction errors fell between 0.5 and 2 for 100% of the children with CKD G2 and CKD G3 and 83% of the children with normal kidney function. Mean prediction errors per CKD group were all within the predefined target range of 0.5- to 2-fold, and model performance was not age dependent (Appendix A).

### 3.3. Prospective Simulations of 3TC and FTC PK in Paediatric CKD Populations 

Administration of the standard dose of 3TC and FTC to adult and paediatric virtual CKD populations resulted in comparable proportional increases in AUC_0–24h_ and C_max_ with advancing disease (Appendix A). For 3TC and FTC, the absolute AUC_0–24h_ was higher in paediatric subjects with normal kidney function and CKD compared to the respective adult populations (Figure 3).

Figure 4 shows the simulated plasma concentration–time curves for healthy adults and children receiving the standard dose and adults and children with various degrees of CKD, receiving a GFR-adjusted dose of 3TC or FTC. Virtual paediatric subjects with CKD G3 and G4 who received a GFR-adjusted 3TC dose had a 1.15- and 1.23-fold higher mean AUC_0–24h_, respectively, compared to those with normal kidney function receiving the standard dose (Figure 3). In adults, the AUC_0–24h_ of 3TC increased by 1.30- (CKD G3) and 1.35-fold (CKD G4), as can be seen in Appendix A. Simulations of FTC in paediatrics showed a 1.20- and 1.30-fold increase in AUC_0–24h_ after GFR-adjusted doses for CKD G3 and G4, respectively, compared to subjects with normal kidney function receiving standard doses. In adults with CKD G3 and G4, FTC AUC_0–24h_ increased 1.44- and 1.52-fold, respectively (Appendix A).

## 4. Discussion

To our knowledge, we are the first to develop a semi-mechanistic paediatric CKD PBPK model in Simcyp^®^ using the Mech KiM module. The paediatric CKD population models were verified successfully using GCV as a surrogate compound and subsequently applied to prospectively simulate PK of 3TC and FTC in paediatric subjects with CKD. Upon administration of the standard dose of 3TC and FTC, exposure appeared to be substantially higher in virtual children with CKD G3 and CKD G4 compared to those with normal kidney function. Simulations of GFR-adjusted dosages for children, as reported in the product label for 3TC or extrapolated from recommendations for adults with CKD for FTC, resulted in slightly increased, yet adequate, exposure for both drugs. These findings support the use of GFR-adjusted dosages for children with CKD, as the risk of additional toxicity is believed to be limited with the 15–30% increased exposure seen in our simulations, outweighing the risk of underdosing when further reducing the GFR-adjusted dosages.

In this study, we demonstrated that PBPK modelling might be a useful tool to predict the PK of compounds that are renally cleared in paediatric subjects with CKD. Other studies have successfully used PBPK modelling to simulate PK of renally excreted drugs in paediatric CKD populations by solely reducing the GFR of the virtual population, as compounds included in these studies did not undergo tubular secretion like 3TC and FTC [37,38,39,40]. Ye et al. applied PBPK modelling to predict PK of ertapenem in paediatric CKD populations, considering both glomerular filtration and tubular secretion [41]. As their population model did not allow for mechanistic scaling of kidney parameters, an additional pragmatic clearance component was included to correct for a reduction in tubular secretion in children with CKD. In contrast, we adopted a more mechanistic approach by reducing kidney size and PTCPGK, similar to previous PBPK studies with adult CKD populations [27,28,42]. 

The finding that the AUC_0–24h_ of both 3TC and FTC, upon standard dose, is slightly higher in the paediatric population without CKD compared to the adult population without can be explained by the difference in weight-based dose. Adults receive 2.9 mg/kg FTC and 4.3 mg/kg 3TC, based on an average weight of 70 kg, which is approximately half of the paediatric dose (6 mg/kg and 10 mg/kg, respectively). For 3TC, this is already confirmed, and dose adjustments have been proposed for several age and weight groups^49^. These differences in doses also explain the higher paediatric AUC_0–24h_ compared to the adult AUC_0–24h_ in virtual CKD subjects receiving the GFR-adjusted dose. 

We also noticed a slightly increased AUC_0–24h_ comparing subjects with CKD receiving GFR-adjusted dosages to subjects without CKD receiving standard dosages. These findings are consistent with PK data observed in adults with and without CKD [20]. Therefore, taking into account the low chance of toxicity related to high doses, we assume that the suggested GFR-adjusted dosages for children with CKD are adequate. Additionally, the AUC_0–24h_ ratio (CKD/non-CKD) was similar across all age groups for 3TC and FTC, in accordance with other PBPK studies that have examined renally excreted drugs in children with CKD [39,43]. Furthermore, a population PK study found a comparable effect of kidney function reduction on the clearance of ceftriaxone for children versus adults, a renally excreted antibiotic drug [44]. 

It is essential to note that no relationship was established between 3TC and FTC plasma concentrations and their efficacy or toxicity, presumably because both NRTIs require intracellular phosphorylation to become active [45]. However, a trend towards an increased risk of neutropenia was seen in adults receiving a 3TC dose of 20 mg/kg/day compared to those receiving 4 mg/kg/day, the latter being the adult standard dose of 300 mg [46]. For adults, a retrospective cohort study showed that the standard dose of 3TC in patients with CKD G3 is well tolerated [47]. Furthermore, experts suggest that this may also apply to FTC for adult patients with CKD [48]. However, when comparing the observed 3TC AUC_0–24h_ upon a 20 mg/kg dose in adults reported by Pluda et al. with an AUC_0–24h_ upon a standard paediatric dose administered to CKD G3 virtual paediatric subjects, the AUC in paediatrics was found to be 1.4-fold higher than that in adults, and it is, therefore, questionable whether the standard dose for paediatric subjects with CDK G3 is safe [46]. This is particularly concerning in the presence of other medications that are often used by children with HIV and could interact with 3TC through inhibition of OCT2 and MATE1, e.g., cotrimoxazole, as this could result in an even further increase in 3TC exposure [49]. For FTC, the simulated C_max_ for children with CKD aged 6–12 years and receiving GFR-adjusted doses of FTC capsules with extended dosing interval seemed higher compared to adults (for CKD G3 and CKD G4 4.04 and 4.93 mg/L and for adults 2.71 en 3.30 mg/L, respectively). Although no clear relationship between PK and toxicity has been established, it is unclear whether a high C_max_ could result in an increased risk of toxicity as higher doses were not studied. In addition, it is important to consider that the exposure–safety relationship of drugs may be age-dependent, as children may be more sensitive to certain side effects compared to adults [50,51].

The prerequisite for using GFR-adjusted dosages is the ability to measure kidney function. In many HIV-endemic settings, however, GFR measurements are not regularly conducted in children living with HIV. As a result, all children generally receive the standard dose of 3TC or FTC. HIV-associated kidney disease in children generally progresses to CKD, resulting in potential chronic overexposure to 3TC or FTC for those receiving the standard dose [6]. Even if a dose adjustment in paediatric patients with CKD is desired, the use of fixed-dose combinations and limited availability of separate formulations in resource-limited settings complicate this in clinical practice [14,47]. Abacavir, zidovudine, and tenofovir alafenamide (drugs included in the fixed-dose combinations with 3TC or FTC) do not require dose adjustments based on kidney function [14]. This concern requires a careful evaluation of the importance of adequate 3TC and FTC dosing with the risks of overdosing the co-administered drugs, while also considering widespread accessibility and harmonization of cART.

This study has several limitations, and some assumptions were made to allow for modelling in paediatric CKD populations.

Our method assumes that a reduction in glomerular filtration is accompanied by a decrease in tubular secretion and reabsorption due to a loss of function of the entire nephron [52]. We used previously established parameters for PTCPGK and CL_add_ to pragmatically apply a reduction in tubular secretion and CL_add_. As GFR cut-off values to classify CKD groups differed between the studies from which these parameters were derived and our intended CKD groups, parameters were calculated using a linear regression formula, established based on available data. However, it should be noted that the relationship between GFR and tubular secretion reduction is non-linear for many drugs, which complicates accurate mechanistic PBPK modelling [19]. The only currently available method to correct for different GFR vs. tubular secretion reduction relationships is by using drug-specific reductions in PTCPGK or individual transporter activity [27]. It should also be noted that while the effects of age on transporter activity and population variability were incorporated in the Mech KiM, the absence of CKD data precluded the ability to integrate any potential effect of CKD on transporter activity [53,54]. Since model verifications for adults and children with CKD for GCV were successful, we assumed that this effect is negligible.

Due to a lack of clinical PK data of 3TC and FTC in children with CKD, GCV was selected as a surrogate compound for validation of the paediatric CKD models. As clinical GCV data in children with CKD G4 (GFR 15–30 mL/min/1.73 m^2^) was absent, we were unable to verify the paediatric CKD model for this population. Additionally, we assumed that, in addition to filtration and tubular secretion, CL_add_ is also decreased in paediatric subjects with CKD, similar to adults. While CL_add_ parameterization appeared to be adequate for predicting 3TC and FTC PK in adults (based on successful model verifications), we were not able to confirm whether it was reasonable to assume a decrease in CL_add_ in paediatric subjects with CKD, because the compound model for GCV did not include an additional clearance component. It should be noted that CL_add_ is a non-mechanistic route of elimination and could potentially result in bias when extrapolating to other populations. The CL_add_ component of the 3TC and FTC compound models may represent different renal and non-renal routes of elimination, i.e., a larger part of FTC is metabolized prior to elimination than 3TC (about 14% versus 5.2%, respectively) [55,56]. We, however, assumed that the proportional decrease in FTC CL_add_ for CKD populations is similar to 3TC, and this approach resulted in adequate predictions of FTC PK. 

The adult and paediatric CKD population models were developed based on the characteristics of a healthy white Caucasian population. However, the majority of the data used for verification of the models for 3TC and FTC was obtained from individuals living with HIV, and the children included in the clinical trials were primarily living in sub-Saharan Africa. It is important to note that HIV infection and ethnicity may impact PK of these compounds [57,58]. Furthermore, in our PBPK model, we assumed that there are no HIV-specific aspects characterizing CKD and, hence, the “general” adult CKD population models are used. Additionally, it is assumed that the mechanisms underlying CKD in children are comparable to those in adults with CKD. This assumption may not hold true in reality, potentially leading to unanticipated differences in renal clearance of 3TC and FTC in CKD children compared to CKD adults.

In order to enhance the utility of PBPK modelling for special paediatric populations, such as HIV and/or CKD, it is essential to acquire additional pathophysiological data. Knowledge on, for instance, the effect of CKD on kidney transporter activity is essential for adequate mechanistic PBPK model parameterization. A better understanding of the unique characteristics of both HIV and CKD and their potential effects on PK and, hence, drug dosing is warranted in order to implement PBPK model-informed doses in clinical practice.

In conclusion, we established PBPK population models for children with CKD, which were verified with a compound model for GCV, a drug that is cleared through glomerular filtration as well as tubular secretion. Using these paediatric CKD population models, it was shown that GFR-adjusted dosages of 3TC and FTC result in adequate drug exposure in children with CKD. These findings support the use of GFR-adjusted dosing for 3TC and FTC in this special population. Clinical PK studies or therapeutic drug monitoring studies are needed to confirm whether the GFR-adjusted doses are safe and effective in children living with HIV and CKD receiving 3TC or FTC as part of cART.

## Figures and Tables

**Figure 1 pharmaceutics-15-01424-f001:**
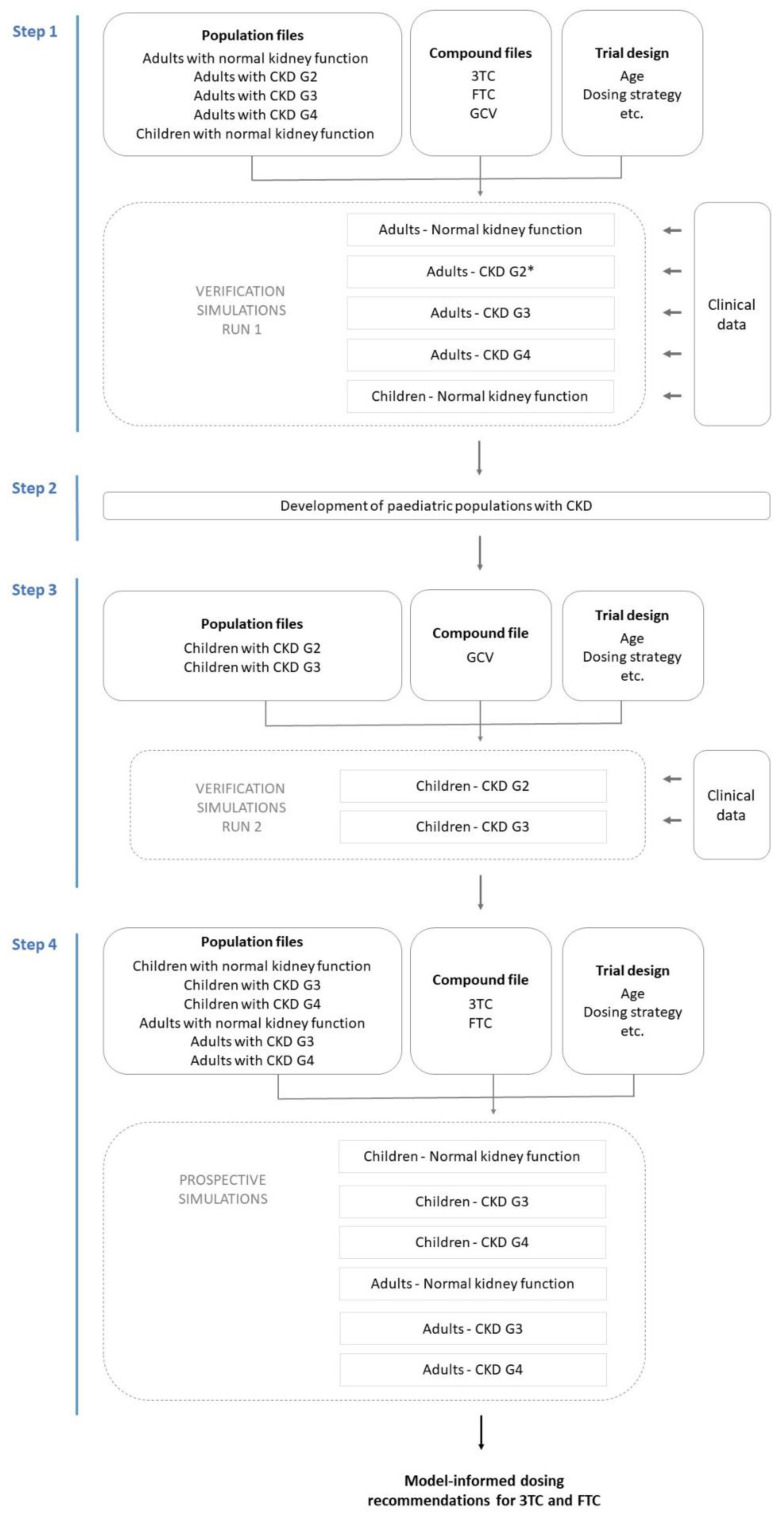
Schematic overview of the stepwise PBPK modelling workflow. In four steps, verification of compound and population models was performed, paediatric CKD population models were established and verified, and final prospective simulations for 3TC and FTC model-informed dose recommendations were conducted. Virtual populations with varying severity of CKD, as presented by grade 2 to 4 (G2–4) are used for simulations. * CKD G2 population only verified for GCV, (not for 3TC and FTC. Abbreviations: 3TC, lamivudine; CKD, chronic kidney disease; GCV, ganciclovir; FTC, emtricitabine.

**Figure 2 pharmaceutics-15-01424-f002:**
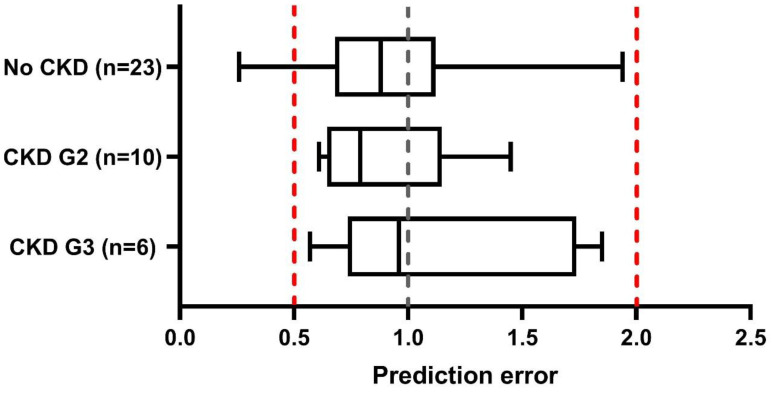
Box whisker plots for the GCV AUC prediction errors describing paediatric CKD model performance. Prediction errors (predicted/observed AUC values) are shown for individual subjects, using observed GVC PK data. The whisker plots reflect the minimum, IQR25, median, IQR75, and maximum prediction errors per group. Results are combined per paediatric CKD population group; no CKD includes paediatric subjects with a GFR of >90 mL/min/1.73 m^2^ (n = 23), CKD G2 between 60 and 90 mL/min/1.73 m^2^ (n = 10), and CKD G3 between 30 and 60 mL/min/1.73 m^2^ (n = 6). The black dashed line represents a prediction error of 1.0 (unity line) and red dashed lines indicate the prediction error range of 0.5 to 2. Abbreviations: CKD, chronic kidney disease; GVC, ganciclovir.

**Figure 3 pharmaceutics-15-01424-f003:**
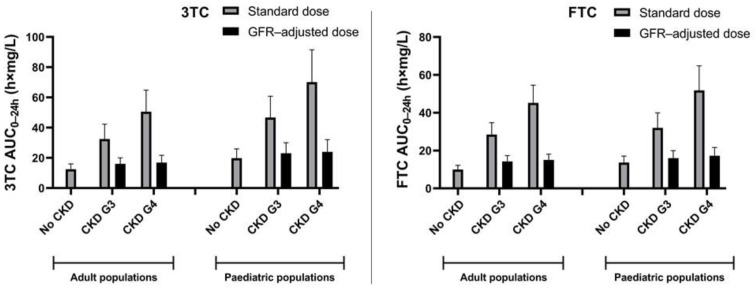
Simulated AUC_0–24h_ for paediatric and adult populations without CKD and with various degrees of CKD receiving either a standard dose or a GFR-adjusted dose of 3TC or FTC at steady-state conditions. **Left panel**: Simulated AUC_0–24h_ of 3TC. Given that the dosing interval for children aged 3 months to 2 years was 12 h, we multiplied the AUC_0–12h_ by a factor 2 to calculate an AUC_0–24h_. **Right panel**: Simulated AUC_0–24h_ of FTC. Given that the dosing interval for simulated populations using FTC capsules was 48 h (CKD G3) or 72 h (CKD G4), we divided the AUC_0–τ_ by a factor 2 or 3 to calculate an AUC_0–24h_, respectively. Data are depicted as population geometric means ± SD.

**Figure 4 pharmaceutics-15-01424-f004:**
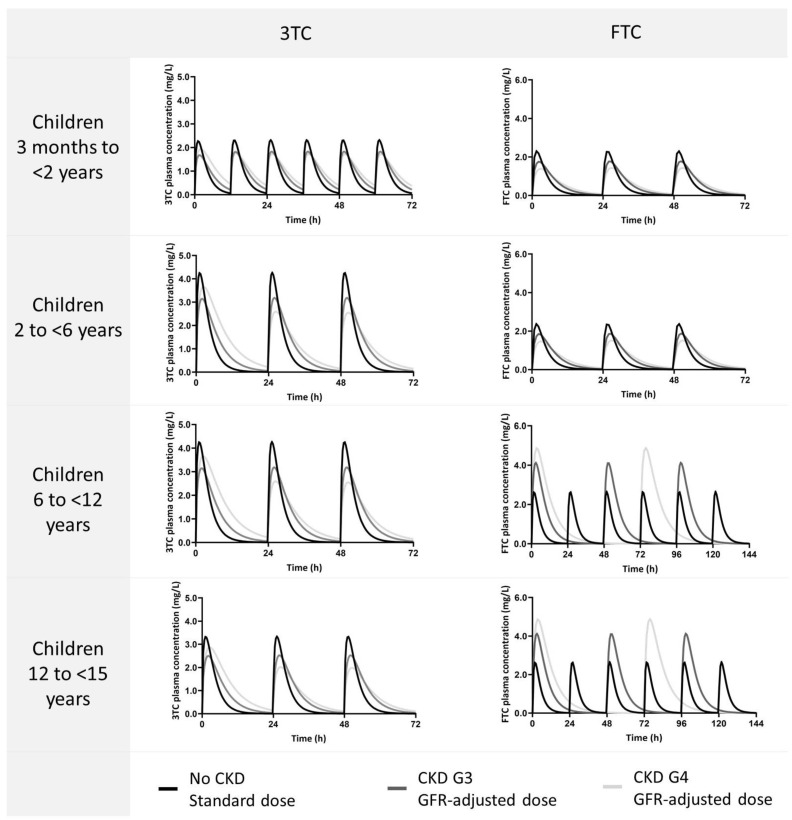
Simulated plasma concentration–time curves in virtual paediatric populations without CKD and various degrees of CKD receiving standard and GFR-adjusted dosages. Dosing strategies are provided in Table 2. Data are presented as population means.

**Table 1 pharmaceutics-15-01424-t001:** Development of adult and paediatric CKD population models.

	Adult CKD Models	Paediatric CKD Models
**CL_filt_**	**GFR** cut-off values (i.e., 30–50 mL/min/1.73 m^2^) were set to reflect the GFR in the various CKD populations *.	In order to define various paediatric CKD populations, we pragmatically included scaling factors in the validated Simcyp^®^ default GFR equation to create paediatric CKD population models with specified **GFR** ranges.
**CL_tub_**	**Kidney size** is adjusted for each CKD group, by reducing kidney size baseline (the equation offset), body weight and body height coefficients *.	Paediatric individual **kidney size** calculation is solely based on body weight. Hence, we have reduced kidney size in paediatric populations by calculating the ratio for body weight coefficients (CKD/no CKD) in adults and incorporated these ratios in the equation determining the paediatric kidney size.
	For 3TC and GCV, drug-specific **PTCPGK** reductions in specific CKD populations (defined differently than in this study) have been set and verified previously [25,27]. The established correlations between GFR and PTCPGK were used to determine PTCPGK values for the adult CKD populations in this study. In the absence of quantitative data for FTC model parameterization, it was assumed that the PTCPGK for the different CKD groups was similar for FTC as for 3TC.	**PTCPGK** for GCV, 3TC, and FTC was reduced according to the reduction applied in the adult CKD population models.
**CL_add_**	There is no **CL_add_** component in the GCV compound model and hence CL_add_ is not adjusted for CKD populations. For 3TC, CL_add_ reductions in specific CKD populations (defined differently than in this study) have been set and verified previously [25]. The established correlations between GFR and CL_add_ were used to determine CL_add_ values for the adult CKD populations in this study. The effect of CKD on CL_add_ was assumed to be similar for FTC and 3TC. Hence, a similar proportional reduction was used for FTC CL_add_ as compared to the 3TC CL_add_ reductions.	**CL_add_** for 3TC and FTC was reduced according to the reduction applied in the adult CKD population models. A CL_add_ component was not included in the GCV model and hence not adjusted in case of CKD.

Renal filtration (CL_filt_), tubular secretion (CL_tub_), and additional clearance (CL_add_) were adjusted to reflect the in vivo paediatric populations with CKD. It should be noted that the CL_filt_ was yet adjusted in the default adult CKD population models. The model parameters that have been adjusted are indicated in bold. * Default setting in Simcyp^®^ adult CKD population models. Abbreviations: 3TC, lamivudine; CKD, chronic kidney disease; GCV, ganciclovir; GFR, glomerular filtration rate; FTC, emtricitabine; PTCPGK, proximal tubule cells per gram of kidney.

**Table 2 pharmaceutics-15-01424-t002:** Dosing strategies of 3TC and FTC, based on their respective product labels, used for prospective simulations of PK in paediatric and adult subjects with different degrees of CKD.

Drug and Formulation	Age of Simulated Subjects	Population and Dose (Standard Dose or GFR-Adjusted Dose)		
		No CKD	CKD G3 (GFR 30–50 mL/min/1.73 m^2^)	CKD G4 (GFR 15–30 mL/min/1.73 m^2^)
**3TC** Dispersible tablets/oral solution	3 mo to <2 yrs	Standard dose: 5 mg/kg/12 h		
		Standard dose	2.5 mg/kg/12 h	2.5 mg/kg LD, 1.88 mg/kg/12 h MD
	2 to <6 yrs	Standard dose: 10 mg/kg/24 h		
		Standard dose	5 mg/kg/24 h	5 mg/kg LD, 3.25 mg/kg/24 h MD
**3TC** Film-coated tablets	6 to <12 yrs	Standard dose: 300 mg/24 h		
		Standard dose	150 mg/24 h	150 mg LD, 100 mg/24 h MD
	12 to <15 yrs	Standard dose: 300 mg/24 h		
		Standard dose	150 mg/24 h	150 mg LD, 100 mg/24 h MD
	≥15 yrs	Standard dose: 300 mg/24 h		
		Standard dose	150 mg/24 h	150 mg LD, 100 mg/24 h MD
**FTC** Oral solution	3 mo to <2 yrs	Standard dose: 6 mg/kg/24 h		
		Standard dose	3 mg/kg/24 h ^#^	2 mg/kg/24 h ^#^
	2 to <6 yrs	Standard dose: 6 mg/kg/24 h		
		Standard dose	3 mg/kg/24 h ^#^	2 mg/kg/24 h ^#^
**FTC** Capsules	6 to <12 yrs	Standard dose: 200 mg/24 h		
		Standard dose	200 mg/48 h ^#^	200 mg/72 h ^#^
	12 to <15 yrs	Standard dose: 200 mg/24 h		
		Standard dose	200 mg/48 h ^#^	200 mg/72 h ^#^
	≥18 yrs	Standard dose: 200 mg/24 h		
		Standard dose	200 mg/48 h	200 mg/72 h

Simulations with both the standard and GFR-adjusted doses were conducted, in both adults and paediatric populations, with no CKD, CKD G3 and CKD G4. ‘Standard dose’: dosing recommendations from the label for patients with normal kidney function, ‘GFR-adjusted dose’: dose recommendation from the label (for 3TC and FTC for adults and paediatrics weighing at least 33 kg with CKD and for 3TC for paediatrics with CKD) or scaled using a similar proportional dose reduction (FTC for paediatrics with CKD) for patients with CKD. ^#^ As no FTC dose recommendations for children are included in the product label, the GFR-adjusted doses were based on FTC dosing recommendations for adults with CKD. A similar proportional dose reduction has been applied for those using oral solution and a similar prolonged dosing interval for those using capsules. Abbreviations: 3TC, lamivudine; CKD, chronic kidney diseases; FDA, US Food and Drug Administration; FTC, emtricitabine; G, grade, GFR, glomerular filtration rate; LD, loading dose; MD, maintenance dose; mo, months; yrs, years.

## Data Availability

The data presented in this study are available on request from the corresponding author.

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
