# Peer review of "Lamivudine and Emtricitabine Dosing Proposal for Children with HIV and Chronic Kidney Disease, Supported by Physiologically Based Pharmacokinetic Modelling"

_pharmaceutics, 2023, doi:10.3390/pharmaceutics15051424_

Round 1

Reviewer 1 Report

The article was accepted because is a good PK model and validated

Giacomo

Author Response

We thank the reviewer for the positive feedback.

Reviewer 2 Report

The manuscript describes PBPK modelling approach to propose lamivudine and emtricitabine dosages for children with HIV infection and renal impairment. The study is well designed and the results clearly presented. The main drawback is a lack of clinical data to confirm the validity of  GFR-adjusted dosing strategies proposed for both drugs.  In addition, both anti-HIV drugs are usually administered in combinations with other drugs. This may lead to pharmacokinetic drug–drug interactions. The proposed   models and simulations did not take this into account. The effects of genetic polymorphism of the relevant drug transporters should be discussed or incorporated into the models.

Author Response

Thank you for your comments. We have adressed them: 

1. We do not expect clinically significant interactions with other antiretroviral drugs. However, cotrimoxazole may interact with lamivudine through inhibition of OCT2 and MATE1, leading to non-clinical significant increase in 3TC exposure. This consideration has now been added to the discussion. See Line 391-394. "This is particularly concerning in presence of other medications that are often used by children with HIV and could interact with 3TC through inhibition of OCT2 and MATE1, e.g. cotrimoxazole, as this could result in even further increase of 3TC exposure[49]."

2. Concerning genetic poplymorphism of the drug transporters, this has been taken into account in the population model as part of the general variability in transporter activity, see Line 428. "It should also be noted that while the effect of age on transporter activity and population variability were incorporated in the Mech KiM, the absence of CKD data precluded the ability to integrate any potential effect of CKD on transporter activity[53,54]."

Reviewer 3 Report

This is a regular PBPK manuscript, nothing new or innovative. This type of PBPK manuscripts are regularly published in wide variety of journals. The finding of this model-based work will require validation from clinical studies before any kind of implementation in a real clinical world.

I have no comment on this manuscript.

Author Response

We thank the reviewer for the feedback. Of note, the consideration with regards to clinical data needed to confirm our findings was also included in the Conclusion: 'Clinical PK studies or therapeutic drug monitoring studies are needed to confirm whether the GFR-adjusted doses are safe and effective in children living with HIV and CKD receiving 3TC or FTC as part of cART.'

Reviewer 4 Report

From a biostats and clinical epidemiology point of view, I wish to warmly congratulate the Authors for their manuscript. Some comments from mine:

- the use of ganciclovir as a proxy for 3TC and FTC has been clearly stated, but its rationale is quite lacking, you should go more in depth for this topic

-  the use of Simcyp for PBPK modeling has been an optimal choice, since this environment is nowadays an intl standard in PK simulation research

- likewise, the proposed modeling flow is pretty adequate (table 1)

- lines 213-235-250-271-335 "Error! Reference source not found", there're some typos to be solved

- Supplements are very informative, but I'm unable to understand why you have added a second references list, IMHO it should be only one!

Author Response

Thank you for your comments. We have adressed them:

- the use of ganciclovir as a proxy for 3TC and FTC has been clearly stated, but its rationale is quite lacking, you should go more in depth for this topic

The rationale for using ganciclovir as a proxy is discussed in the Materials and Methods section. We have now provided some additional details, see Line 103-104. "Similar to 3TC and FTC, GCV undergoes both glomerular filtration, in similar proportions as compared to 3TC and FTC, and tubular secretion (through Organic anion transporter 1 (OAT1) and MATE) and clinical PK data are available for this drug in children with CKD."

-  the use of Simcyp for PBPK modeling has been an optimal choice, since this environment is nowadays an intl standard in PK simulation research - likewise, the proposed modeling flow is pretty adequate (table 1)

This has been noted with thanks.

- lines 213-235-250-271-335 "Error! Reference source not found", there're some typos to be solved

These errors have been solved, thanks.

- Supplements are very informative, but I'm unable to understand why you have added a second references list, IMHO it should be only one!

To our understanding, the supplementary files should have a separate reference list. @editor: is this correct or should we adjust the reference list in the supplementary file?